# Coeditor: Leveraging Repo-level Diffs for Code Auto-editing

**Jiayi Wei**[*]
Augment Computing, Inc.
jiayi@augmentcode.com

**Greg Durrett, Isil Dillig**
University of Texas at Austin
{gdurrett, isil}@cs.utexas.edu

## Abstract

Developers often dedicate significant time to maintaining and refactoring existing code. However, most prior work on generative models for code focuses solely on creating new code, overlooking the distinctive needs of editing existing code. In this work, we explore a multi-round code auto-editing setting, aiming to predict edits to a code region based on recent changes within the same codebase. Our model, Coeditor, is a fine-tuned language model specifically designed for code editing tasks. We represent code changes using a line diff format and employ static analysis to form large customized model contexts, ensuring the availability of appropriate information for prediction. We collect a code editing dataset from the commit histories of 1650 open-source Python projects for training and evaluation. In a simplified single-round, single-edit task, Coeditor significantly outperforms GPT-3.5 and SOTA open-source code completion models (bringing exact-match accuracy from 34.7 up to 60.4), demonstrating the benefits of incorporating editing history for code completion. In a multi-round, multi-edit setting, we observe substantial gains by iteratively conditioning on additional user edits. We have open-sourced our code, data, and model weights to encourage future research and have released a VSCode extension powered by our model for interactive IDE usage.

## 1 Introduction

In recent years, there has been enormous interest in applying transformer models for code generation (Feng et al., 2020; Ahmad et al., 2021; Wang et al., 2021; Chen et al., 2021; Fried et al., 2022; Allal et al., 2023), which has led to impressive performance on tasks such as program synthesis (Li et al., 2022; Nijkamp et al., 2022), program translation (Lachaux et al., 2020; Szafraniec et al., 2022), type inference (Jesse et al., 2022; Wei et al., 2023), and code auto-completion (Guo et al., 2021; Svyatkovskiy et al., 2021; Nguyen & Nadi, 2022; Zhang et al., 2023).

While these approaches effectively help programmers *creating* new code, they are not as adept at assisting with *revising* existing code. Code completion tools like GitHub Copilot do not track programmers' changes and cannot predict where and how to make additional modifications. However, during a software project's development cycle, developers often spend significant time editing code—changes made to one part of the codebase typically affect many others, and manually propagating these changes can be tedious and time-consuming.

In this paper, we introduce a task that we call (multi-round) *auto-editing* where the goal is to predict edits to code conditioned on the user's previous edits. In particular, given an original codebase $U$ and a set of code changes $\Delta_1, \ldots, \Delta_k$ that are semantically related (like those forming part of a commit), the auto-editing problem is to predict how to modify a specified region of code $u \in U$ by learning the following distribution:

$$P(\Delta u \mid \Delta_k \ldots \Delta_1, U) . \tag{1}$$

Importantly, we allow the target region $u$ to overlap with any previous modifications $\Delta_1, \ldots, \Delta_k$ to support repeated editing of the same region. This formulation enables the workflow illustrated in Figure 1, where a user can work alongside the model in multiple editing rounds, accepting suggestions matching the user's intent and making additional edits manually if necessary.

---

[*]Work performed while at UT Austin.

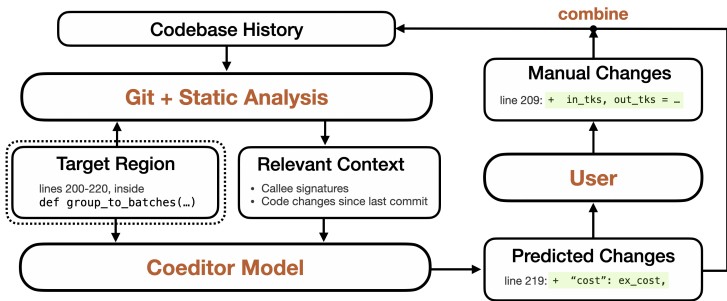

Figure 1: The multi-round auto-editing task. The user inspects the model output in each editing round and can optionally perform manual editing.

To solve this problem, we propose a new model called Coeditor that builds on top of the established CodeT5 model architecture and pre-trained checkpoint (Wang et al., 2021). Coeditor is based on two key ideas. First, it encodes all prior code edits $\Delta_1, \ldots, \Delta_k$ using a line-based diffing scheme and decodes $\Delta u$ using masked span infilling; and, second, it uses lightweight static analysis to pull in relevant parts of the codebase $U$. To effectively handle large contexts with numerous code changes, we also replace CodeT5's dense attention with a block-sparse attention pattern, allowing us to reduce the computational cost while maintaining the ability to attend to all relevant code changes.

Another challenge in developing Coeditor is the lack of suitable training data for multi-round auto-editing. We address this issue by collecting a new dataset, PYCOMMITS, from the commit histories of 1650 open-source Python projects on GitHub. We compute tree-differences between adjacent codebase versions to identify modifications to the same Python function and randomly split some changes into the model input for training in repeated editing scenarios. During testing, we use ground truth code changes to simulate user decisions regarding when to accept partial changes suggested by the model and when to manually perform edits missed by the model.

We compare our approach against existing code infilling models and show that, even in a simplified setting that requires predicting a *single* edited line in isolation, they severely lag behind our change-aware model: our method achieves 60.4% exact match accuracy using a 220M parameter model, whereas `text-davinci-003`—the best performing code infilling model we have compared, which has 175 billion parameters—achieves only 30.5%. In the full multi-round setting, we found that Coeditor automates editing 46.7% of the changed lines, saving the user 28.6% of keystrokes measured by an edit distance metric that accounts for cursor movement.

In summary, this paper presents the following main contributions:

- We introduce the repo-level multi-round code editing task, along with the corresponding PYCOMMITS dataset and evaluation framework.

- We introduce a new code editing model derived from CodeT5, using a line diff-based encoding scheme and enhancements that enable the model to condition on long contexts and appropriate other parts of the codebase, addressing key challenges in this setting.

- We release our source code, dataset, model checkpoint, as well as a VSCode extension that supports interactive usage to foster future research.[1]

## 2 MOTIVATING EXAMPLE

In this section, we illustrate our technique using the example in Figure 2, showcasing a two-round interaction between the user and our Coeditor model. Subfigures (a) and (b) display two initial user changes, while subfigures (c) and (d) illustrate two sequential Coeditor invocations with inlined model suggestions. We further analyze this example in detail below.

First, the user modifies the `pack_batch` function in subfigure (a) to read a new dictionary key, "`cost`", from each row in the input. The extracted values are used to compute the total cost

---

[1]Available at https://github.com/mrvplusone/Coeditor.

Figure 2: An example usage of Coeditor. (a) The user first edits the `pack_batch` function to read an additional dictionary key, "`cost`", from each row in the input. (b) The user then removes 3 lines at the top of the `group_to_batches` function. (c) The user now invokes Coeditor at the bottom half of the same function. Coeditor correctly suggests adding a "`cost`" key to the dictionary variable `row`, but it fails to address the now undefined variables underlined in red. (d) However, if the user accepts the suggested change and manually introduces two new variables at line 209, Coeditor can then suggest the correct changes accordingly.

of the batch and added to the output. Next, the user removes three lines at the top of the `group_to_batches` function in subfigure (b). By removing these three lines, the user wants to avoid creating these lists beforehand and instead plans to call the `process_edit` function inside the for loop below.

The user then scrolls down and invokes Coeditor at the bottom half of the same function (subfigure c). Here, the modified `pack_batch` function is called at lines 225 and 228 in subfigure (c), and its argument `current_batch` is iteratively constructed from `row`, which is a dictionary defined at line 215. Hence, the model correctly infers that `row` should be updated to include a "`cost`" key. Examining the surrounding context, the model also identifies that the `ex_cost` variable (defined at line 209) should be used as the inserted dictionary value.[2]

While Coeditor makes some useful editing suggestions so far, it does not address the now-undefined variables underlined in red by the IDE in subfigure (c). In particular, as there are no obvious alternatives nearby to replace these variables, Coeditor is unable to automatically fix these errors. Such a situation is common when the surrounding changes alone do not provide sufficient information to derive a complete solution. Therefore, the user can accept the partial changes suggested by the model and then manually introduce two new variables at line 209, as shown in subfigure (d). Coeditor can then leverage these new variables to suggest the correct changes needed to fix the errors.

This iterative approach allows Coeditor to adapt and refine its suggestions based on additional user edits, providing a more effective code editing experience than existing code completion techniques. By incorporating the editing history into the prediction context, Coeditor can better assist developers in a wide range of code editing tasks, from simple modifications and refactoring to more complex

---

[2]Coeditor produces the same result even when `pack_batch` is defined far away or in a different file, as Coeditor tracks all changes the user has made since the last commit and incorporates them into its context.

codebase-wide updates. For a more comprehensive understanding, we have included a screen recording of the Coeditor VSCode Extension, which can be found in the supplementary material.

## 3 METHODS

Recall from the introduction that we wish to model the distribution $P(\Delta u \mid \Delta_k \ldots \Delta_1, U)$. To this end, we first describe how to encode the target change $\Delta u$ and contextual changes $\Delta_1 \ldots \Delta_k$ (subsection 3.1). We then describe how to form the context from the codebase $U$ using function signatures (subsection 3.2). These choices naturally lead to a model compatible with fine-tuning CodeT5, which was pre-trained on the masked span infilling task (subsection 3.3). Finally, we describe our new dataset that is used to fine-tune this model (subsection 3.4).

### 3.1 ENCODING CODE CHANGES

A suitable format is required to map code changes into token sequences that can be processed by a seq2seq transformer language model. In our setting, we want to select a format that encodes and decodes code changes in a uniform manner while minimizing the number of tokens the model needs to produce. Hence, we adopt a line-diff-based format, enabling us to convert auto-editing into a masked span infilling problem (Wang et al., 2021).[3]

Consider a block of code $u$ to be made up of lines $l_1, \ldots, l_m$ and a user-specified edit region that spans between line $a$ and $a + n$, where $1 \le a \le a + n \le m$. Moreover, each line is associated with a status variable $s_i$ indicating what type of change (if any) has already been made; $s_i \in \{\texttt{(empty)}, \texttt{<add>}, \texttt{}\}$.[4] We encode the input code by a function EncInput that (optionally) prepends status tokens $s_1 \ldots s_m$ and placeholder tokens $\texttt{<1>} \ldots \texttt{<n>}$ at the start of each line:

$$\text{EncInput}(u) = s_1 l_1 s_2 l_2 \ldots \texttt{<1>} s_a l_a \texttt{<2>} s_{a+1} l_{a+1} \ldots \texttt{<n>} s_{a+n} l_{a+n} \ldots s_m l_m \ .$$

For contextual changes $\Delta_1 \ldots \Delta_k$, we can encode them using the same format but with an empty edit region. When the target change $\Delta u$ contains line additions, denoting the $j$th line to be inserted before line $i$ as $l'_{ij}$, we can encode $\Delta u$ using the following expression,

$$\text{EncOutput}(\Delta u) = \texttt{<1>} I_a D_a \texttt{<2>} I_{a+1} D_{a+1} \ \ldots \ \texttt{<n>} I_{a+n} D_{a+n} \ ,$$
$$\text{where } I_i = \texttt{<add>} \ l'_{i1} \texttt{<add>} \ l'_{i2} \ \ldots \ \texttt{<add>} \ l'_{i|I_i|} \ ,$$
$$D_i = \text{if } l_i \textit{ is to be deleted} \text{ then } \texttt{} \text{ else } \texttt{(empty)} \ .$$

Note that we add a further restriction that forbids $D_i$ from being $\texttt{}$ if $s_i$ is $\texttt{<add>}$ in order to prevent the model from modifying a line that has just been added; we discuss this in more detail in section 6. Figure 3 illustrates this line-diff-based encoding scheme using the example from Figure 2. This format ensures that if we replace the placeholder tokens in the input with the corresponding changes specified in the output sequence, we obtain the total change that combines $u$ and $\Delta u$.

### 3.2 ANALYZING RELEVANT SIGNATURES

Having described how we encode code changes, we must also establish a method for feeding $U$, the remaining codebase, to the model. Simply inputting the entire codebase as is would result in an excessive number of tokens, overwhelming the context. Instead, inspired by the ideas proposed in previous type inference work (Pradel et al., 2020; Wei et al., 2020; 2023), we employ lightweight static analysis to extract the most relevant information into the context, as outlined below.

For each target code region $u$, we analyze its pre-edit code and generate a list of its usages.[5] In the case of a function usage, we retrieve its function signature; for a variable or class member usage, we retrieve the first statement in which it was assigned. We then concatenate all these usages into a single

---

[3]Prior work has proposed various methods to produce code changes. e.g., Zhang et al. (2022) learns the distribution $P(u' \mid u)$ and Reid & Neubig (2022) tags each input token with a label indicating deletion, insertion, or replacement. However, these methods require more copying or tagging, resulting in longer output sequences compared to our approach.

[4]We represent edits as line diffs output by `Differ.compare` using the standard `difflib` library.

[5]We use the Jedi package for this purpose: https://github.com/davidhalter/jedi.

```
[...]                                          <6>
         ref_size_sum = 0                      <7> <add>         query_size=len(input_tks),
         ref_selected = list[TokenSeq]()           
         for ref in all_refs:                  <8> <add>         output_size=len(output_tks),
<1>          if ref_size_sum + len(ref) <= args.max_total_   
<2>              ref_selected.append(ref)      [...]
<3>              ref_size_sum += len(ref)      <12> <add>        "input_tks": input_tks,
<4><add>         input_tks, output_tks = process_edit(edit, ar    
<5>          ex_cost = retrieval_cost_model(   <13> <add>        "output_tks": output_tks,
<6>              ref_size=sum(len(x) for x in ref_selected),    
<7>              query_size=len(input_tks_list[i]),  <14>
<8>              output_size=len(output_tks_list[i]),
<9>          )
<10>         ref_selected.sort(key=lambda x: id2ref_name[id(x)    at: motivating
<11>         row = {                                   TokenSeq = list[Token]
<12>             "input_tks": input_tks_list[i],       pack_batch(rows: list[dict]) -> dict
<13>             "output_tks": output_tks_list[i],     retrieval_cost_model(ref_size: int, query_size:
<14>             "ref_selected": ref_selected,         int, output_size: int) -> float
<15><add>           "cost": ex_cost,
<16>         }                                   at: motivating.BatchArgs
<17>         if ex_cost > cost_limit:               cost_limit() -> float
<18>             warnings.warn("Batch cost limit is too small.   max_query_tks: int = 512
<19>         if ex_cost + current_cost <= cost_limit:            max_ref_tks: int = 512
[...]                                               max_total_ref_tks: int = 50 * 256
                                               [...]
```

Figure 3: Coeditor encoding format. (Left) the input sequence adds placeholder tokens to indicate code region to edit. (Top right) the output sequence specifies further changes at each placeholder token. (Bottom right) relevant signatures are retrieved from the codebase and added to the context. (In this example, the Python module is called `motivating`).

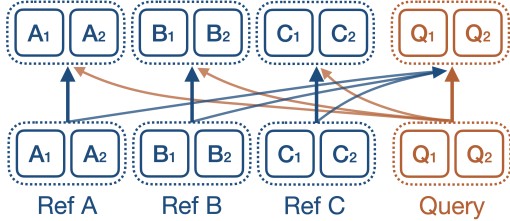

Figure 4: Coeditor encoder sparse attention pattern. All attention between the reference blocks are skipped to avoid the quadratic cost of dense attention.

"document", as shown at the bottom right of Figure 3, which serves as additional input context. This approach allows the model to access the most pertinent information about the current code region and significantly improves model performance (Table 5), while generating only a small number of extra tokens in the context (Table 2).

### 3.3 ADAPTING CODET5

Our model is based on the architecture and pre-trained weights of CodeT5 (Wang et al., 2021). CodeT5 was pre-trained on a large corpus of code data using the masked span infilling objective, making it a suitable choice for our problem. We employ the CodeT5-base model, containing 220M parameters, and fine-tune it for our code auto-editing setting. Although the original CodeT5 model was pre-trained with a small sequence length of 512, its relative positional encoding scheme allows us to fine-tune it on much longer sequences for our problem.

Considering that a single commit may encompass numerous code changes, concatenating all changes into a single input can lead to long token sequences that are difficult for the CodeT5 model to process with dense attention. To mitigate this issue, we replace the full attention in its encoder with a block-sparse attention pattern, illustrated in Figure 4. This pattern divides the input sequences into multiple reference blocks and a query block. The query block contains the code to be edited, whereas each reference block encodes a contextual unit change $\Delta u_j$ or a chunk of the signature document. We limit the sequence length of each block to 512 tokens for references and 1024 for the query, dividing longer blocks into multiple ones if necessary. The self-attention within each block is performed as

Table 1: General statistics of the PYCOMMITS dataset.

|  | train | valid | test |
|---|---|---|---|
| projects | 1550 | 50 | 50 |
| used commits | 217K | 5006 | 5854 |
| modified files | 501K | 10.1K | 11.1K |
| modified functions | 958K | 20.1K | 22.5K |
| modified lines | 7.10M | 143K | 169K |

Table 2: Additional statistics specific to our technique, computed over the test set.

|  | definition | median | mean | max | $\geq$ max |
|---|---|---|---|---|---|
| query tokens | $\mathrm{EncInput}(u)$ | 258 | 361.9 | 1024 | 7.8% |
| output tokens | $\mathrm{EncOutput}(\Delta u)$ | 60 | 89.7 | 512 | 1.3% |
| prev change tokens | $\mathrm{EncInput}(\Delta_1 \ldots \Delta_k)$ | 1625 | 4.14K | 16.4K | 11.2% |
| signature tokens | $\{\mathrm{signature}(v)\}_{v \in \mathrm{usages}(u)}$ | 313 | 515.5 | 15.9K | 0.0% |

usual, but the attention between different reference blocks is skipped to save computation, similar to other retrieval-augmented models (Izacard & Grave, 2021). However, we still allow the query block to attend to and be attended by all reference blocks (a global attention block (Beltagy et al., 2020; Zaheer et al., 2020)). We also set the relative distance between each reference and the query to be infinite when computing the relative positional encoding, making the model is insensitive to the ordering of the references. We are able to use a total of 16.4K reference tokens at test time, which is sufficient to cover 88.8% of problem instances in our test set without truncating the context (Table 2). See subsection A.2 for more discussion of long-document attention mechanisms.

## 3.4 THE PYCOMMITS DATASET

To train our model, we gather real-world code changes from the commit histories of open-source Python projects, a dataset we call PYCOMMITS. For each commit, we first identify which changes are made to the same code unit (a unit can be either a function, a region of a class, or a region of a module) and subsequently separate the commit into a list of unit additions, unit deletions, or unit modifications. As our work primarily focuses on code editing, only unit modifications are used as training labels, while the other two types of changes remain visible to the model as context.

For each unit modification, we create a training problem instance that instructs the model to predict the code change based on all prior (but not future) changes from the same commit. Git does not record the editing order of changes within the same commit, so we employ a simple heuristic that sorts unit changes according to their source code locations and the import order between modules. Specifically, we assume that units within the same file are modified from top to bottom, and if a module imports another module, changes in the imported module occur before those in the importing module.[6] To train our model for the proposed multi-round editing setting, we generate synthetic data demonstrating repeated editing to the same code unit as follows: for those code change involving least two changed lines, we randomly sample a subset of the changes as the prediction target and line the remaining changes into the input. For example, the problem instance shown in Figure 3 can be generated by inlining 2 of the 6 changed lines in the input.

We construct a new code editing dataset using the commit history of 1,650 Python projects with permissive licenses (MIT, Apache, and BSD) sourced from GitHub. We use 50 of the projects for testing and 50 for validation and use the remaining 1,550 projects for training. We use at most 1000 commits per project per project to ensure that the model is trained on a diverse set of code changes. We show the general statistics in Table 1 and the statictics that are specific to our technique in Table 2. Tokenization is performed using the CodeT5 tokenizer.

---

[6]Note that this ordering mainly affects how we generate the training and testing data. At test time, our model can condition on changes both above and below the edit region.

Table 3: Performance on 5000 code completion instances extracted from edits (PYCOMMITS-ONELINE). Add EM and Replace EM are the (enhanced) exact-match accuracies on addition and replacement change, respectively.

| Model | Parameters | Context | Exact Match Rate (%) | | |
|---|---|---|---|---|---|
| | | | Add | Replace | Overall |
| InCoder1B | 1.3B | 2K | 29.0 | 25.2 | 26.2 |
| InCoder6B | 6.7B | 2K | 34.0 | 30.4 | 31.3 |
| SantaCoder | 1.1B | 2K | 31.0 | 28.1 | 28.8 |
| StarCoder7B | 7B | 8K | 37.9 | 33.7 | 34.8 |
| text-davinci-003 | 175B | 4K | 40.2 | 39.3 | 39.5 |
| Coeditor | 220M | 16K | **47.1** | **64.9** | **60.4** |

## 4 EVALUATION

In this section, we first compare Coeditor with prior code completion approaches on a simplified version of the editing task. We then report Coeditor's performance on the proposed multi-round editing task and conduct ablation studies. Example model outputs are included in the appendix.

**Training Setup** We initialize Coeditor with the CodeT5-base checkpoint (220M parameters) and train the model on our training set for 1.75 epoch, gradually increasing the model reference context size from 2048 tokens to 4096 tokens (at epoch 1) and then to 8192 tokens (at epoch 1.5). We use Huggingface's Trainer implementation and the AdamW optimizer, with a linear learning rate schedule with a starting learning rate of 2e-5 and 0.01 weight decay. We train the model with a fixed batch size of 1 and a total of 1.34 million training steps. Training took about 5 days on a single NVIDIA Quadro RTX 8000 GPU with 48 GB memory.

### 4.1 COMPARISON WITH CODE COMPLETION APPROACHES

**Baselines** We compare Coeditor with four open-source code generation models: InCoder-1B, InCoder-6B (Fried et al., 2022), SantaCoder (Allal et al., 2023), and StarCoder7B (Li et al., 2023). All four code generation models have been trained with the Fill-in-the-middle pre-training objective (Aghajanyan et al., 2022). We also compare Coeditor with OpenAI's `text-davinci-003` model, which is the most recent GPT model supporting an infilling interface.[7]

**Creating test instances** We generate code completion problem instances from real commits as follows. For each code change in PYCOMMITS, we take the last changed line as the completion target. If the last change is a modification, we delete the modified line and let the model fill in the new version of the line. If the last change is a deletion, we simply discard the change. We then inline all changes before the target into the prediction context. This inlining process is implemented differently for each model: for our Coeditor model, the inlined changes are visible to our model following the encoding scheme described in subsection 3.1; for the code completion models, we simply apply the inlined changes to the original code and use the resulting state as the model input. Also note that while our model constructs its prediction context using relevant changes and static analysis (as described in subsection 3.2), the code completion models (which are unaware of code changes) only use the code surrounding the completion target as the prediction context. We call this test dataset, derived from our PYCOMMITS test set, PYCOMMITS-ONELINE.

**Results** We report the performance (without fine-tuning on this task) of all approaches in Table 3. We use an enhanced exact-match (EM) accuracy metric that performs semantic-preserving code normalization before checking for string equivalence.[8] All results, except for `text-davinci-003`, were obtained by measuring 5000 randomly sampled code completion problems from our test set. `text-davinci-003` was evaluated using only 1000 problems to reduce cost. We see that Coeditor

---

[7]We also experimented with prompting `gpt3.5-turbo` for this task using either a suffix-then-prefix format or some natural language instructions followed by code with a special `<MISSING LINE>` marker, but these variants worked less well than `text-davinci-003`'s native infilling API.

[8]We normalize Python code by (1) parsing the code into a syntax tree using the `ast` library, (2) removing any comments and docstrings, (3) sorting all keyword arguments in function calls, and (4) un-parsing the syntax tree.

Table 4: Multi-round evaluation results measured on 5000 problems from the PYCOMMITS test set. Lines, Levenshtein, and Keystrokes are the average total gains in the corresponding metrics. Rounds is the average number of rounds needed to complete all desired changes.

| Setting | Lines (%) | Levenshtein (%) | Keystrokes (%) | Rounds |
|---|---|---|---|---|
| SingleRound | 28.5 | 23.1 | 19.2 | 1 |
| MultiRound | 46.7 | 25.9 | 28.6 | 2.43 |

significantly outperforms the other code generation models for both addition and replacement changes. Despite only using a 220M parameter model, Coeditor achieves an overall EM of 60.4%, which is approximately 1.5 times higher than the best performing code completion model (39.5%), demonstrating the significant benefits of incorporating editing history for code completion. We also include 3 example model outputs on this task in the appendix (subsection A.3).

## 4.2 MULTI-ROUND EDITING

This evaluation focuses on the editing assistant use case where we assume the user has some desired code changes in mind, and we aim to evaluate how much the model can save the user's effort by automating as much changes as possible, potentially under the guidance of the user. The user can accept partial changes suggested by the model and make additional changes manually if needed.

**Evaluation workflow**   To evaluate the above use case automatically, we use the ground-truth code changes to simulate the user's actions. In particular, when the model predicts a list of changes, we compare the predicted changes against the ground truth changes line-by-line and accept any line change that exactly matches the ground truth. If none of the suggested changes match the ground truth, we assume the user will manually perform the first remaining change. In both cases, after the additional changes, we rerun the model to obtain new suggestions and repeat until all desired changes have been performed or the round limit = 6 has been reached. In the end, we compute the total gain using the difference between the editing cost of the ground truth and the accumulative editing cost of all manually performed edits.

**Measuring editing cost**   There are multiple ways to measure the cost of performing a code change. Since there is no consensus on the best metric, we report 3 metrics in our results. Prior work (Lavazza et al., 2023) suggests that for code understanding tasks, simple line counts-based metrics are almost as good as more complex metrics, hence our first metric, **Lines**, simply measures the number of changed lines before and after the edit. We also report **Levenshtein**, the classic Levenshtein editing distance metric that measures the minimal number of character addition, deletion, and substitution needed to transform one string into another. Although simple, the Levenshtein distance doesn't model many important aspects of code editing, such as the cost associated with cursor movement, and it also under-count the cost of substitution and over-count the cost of large deletions. Hence, we propose an additional metric, **Keystrokes**, that aims to better approximate the number of needed user keystrokes than Levenshtein by allowing for batch deletion and accounting for the cost of cursor movements. We describe this metric in detail in the subsection A.1.

**Results**   We report the evaluation results on 5000 problems sampled from our test set in Table 4, and we also report the single-round performance for reference. We see that Coeditor achieves much larger total gains under the multi-round setting, especially when measured with the Lines and Keystrokes metric (which we believe more accurately captures the user editing effort than Levenshtein). We also show 3 examples of the model's suggestions in the appendix (subsection A.4).

## 4.3 ABLATION STUDIES

We retrain the model with various components disabled to study their impact on the overall model performance. We report the (single-round) exact match performance of each variation on the entire PYCOMMITS validation set in Table 5. The results show that removing any of the components leads to a decrease in performance, highlighting the importance of each component in the overall model. Specifically, when we remove the explicit feeding of code changes (No Diffs), the EM drops the most, from 42.1% to 26.1%. When we disable the static analysis component (No Signatures), the EM decreases to 33.3%. Using a smaller limit of reference tokens impacts the model performance the

Table 5: Ablation results on the entire validation set (PYCOMMITS). All pairwise differences are statistically significant with $p < 0.05$ using a paired bootstrap test.

| Ablation | Description | EM (%) |
|---|---|---|
| No Diffs | Feeding the same input to the model except that all changes are replaced with their post-edit results alone. | 26.1 |
| No Signatures | Disabling the static analysis component and removing function and class signatures from the prediction context. | 33.3 |
| Small Context | Reducing the max number of reference tokens from 16K to 2048. | 39.8 |
| No Ablation | Model trained with our default settings. | **42.1** |

least, reducing EM to 39.8%. All results reported in Table 5 were obtained by training the model for half amount of training steps to save compute.

## 5 RELATED WORK

The past work most similar to our setting is that of Brody et al. (2020), which also targets a contextual code editing setting. However, it can only predict a restrictive set of code changes expressable as moving, deleting, or copying existing AST nodes and cannot generate novel expressions that are not present in the input. It also doesn't make use of modern transformer architecture or pre-training techniques. In contrast, Ni et al. (2021) takes a rule-based approach, using program synthesis methods to distill similar change patterns in the context and make editing suggestions accordingly.

There is also prior work on non-contextual code change prediction settings. In Chakraborty et al. (2020), the authors use past code patches to train the model to perform similar edits and evaluate it on future edits in the same codebase. However, since the model does not condition on relevant changes, their technique requires retraining the model for new types of edit patterns. Panthaplackel et al. (2020a) proposes augmenting the decoder with a direct copying mechanism to help a encoder-decoder model perform editing tasks. Zhang et al. (2022) proposes a de-noising pre-training scheme, in which they randomly corrupt actual code snippets and train the model to predict the uncorrupted version from the corrupted version. Tufano et al. (2021) focuses on predicting code review changes using developer discussions. Reid & Neubig (2022) focuses on modeling the iterative editing process of texts and code and proposes a different change encoding scheme that represents edits at the word-level based on the Levenshtein algorithm.

Lastly, there is prior work that focuses on on learning to update code comments (Panthaplackel et al., 2020b) or generating natural language descriptions (Panthaplackel et al., 2022) from code changes. This work, we focus on measuring our model's ability to make correct code changes and remove comments and doc-strings before measuring the exact accuracy.

## 6 CONCLUSION AND LIMITATIONS

In this paper, we presented Coeditor, a novel approach for repository-level code auto-editing. Building on the CodeT5 architecture, our model incorporates line diff format and static analysis to create large customized model contexts. We demonstrated that Coeditor significantly outperforms existing code completion methods in a simplified single-round, single-edit task. Furthermore, our model shows substantial performance improvements in a more complex multi-round, multi-edit setting.

One limitation of our current work is assuming that the user would manually identify the region of code requiring changes and then invoke the model to predict where and how to make these changes within that region. A promising direction for future work would be to extend the model to help users identify the regions of code that need to be changed within the entire codebase, enabling new types of usage such as auto-refactoring. Furthermore, as mentioned in subsection 3.1, our model is designed to not modify any lines in the input that have already been modified. While this simplifies the evaluation and prevents undesirable model behaviors such as disregarding changes already made by the user or getting stuck in an infinite editing loop, it can also limit the practical usage of our tool. Therefore, it would be valuable to explore methods that allow the model to collaborate with users in finer-grained editing units in future research.

## ACKNOWLEDGMENTS

This work was conducted in a research group supported by NSF awards CCF-1762299, CCF-1918889, CNS-1908304, CCF-1901376, CNS-2120696, CCF- 2210831, and CCF-2319471. We would like to give special thanks to Miltos Allamanis and Ray Mooney for their invaluable advice on this project. Additionally, we appreciate the constructive comments from the anonymous reviewers, which greatly improved our work.

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

# A  APPENDIX

## A.1  KEYSTROKE DISTANCE

We developed a string distance metric incorporating the cost of cursor movement, approximating the number of keystrokes needed to transform an input string into an output string.

Given the initial state with `i = len(input)`, `j = len(output)`, `cursor_dis = init_cursor_dis`, and `deleting = False`, the cost is calculated using dynamic programming with the optimal combination of the following operations:

- M: Match character (cost=0), requires `input[-i] == output[-j]` and `not deleting`, results in `i -= 1`, `j -= 1`, and `cursor_dis += 1`.

- D: Delete input character (cost=1), requires `cursor_dis == 0` and `not deleting`, results in `i -= 1`.

- A: Add output character (cost=1), requires `cursor_dis == 0` and `not deleting`, results in `j -= 1`.

- C: Move cursor to current position (cost=`min(cursor_dis, cursor_jump_cost)`), requires no conditions, results in `cursor_dis = 0`.

- S: Begin deletion (cost=1), requires `cursor_dis == 0` and `not deleting`, results in `deleting = True`.

- K: Continue deletion (cost=0), requires `deleting`, results in `i -= 1`.

- E: End deletion (cost=1), requires `cursor_dis == 0` and `deleting`, results in `deleting = False`.

Where `cursor_jump_cost` is a constant that we set to 4 when reporting our results. The worst-case complexity of this algorithm is $O(\text{len(input)} \times \text{len(output)} \times \text{cursor\_jump\_cost})$. This model does not consider copying or pasting operations.

Note that in some cases, the total gain can be negative when measured with Levenshtein or Keystroke distance. For example, the Levenshtein distance of modifying a sentence is lower than the total of first deleting the sentence and then adding a new one.

## A.2  DISCUSSION OF SPARSE ATTENTION MECHANISMS

The block-sparse attention pattern we described in subsection 3.3 follows past work on retrieve-and-read models for natural language question answering. Specifically, it resembles Fusion-in-Decoder (Izacard & Grave, 2021) with three changes. First, we have no notion of a question that is jointly encoded with each retrieved snippet. Second, our target code block $u$ is given special status in the encoder and can globally attend to each retrieved snippet. Third, we modify the relative positional encoding to make each query "infinitely" far from the reference.

Our approach also resembles Longformer (Beltagy et al., 2020) or BigBird (Zaheer et al., 2020), most notably in how our query block's cross-attention with the references can be viewed as an instance of global attention. However, our segments do not come from a coherent context, so our local attention component is a block-diagonal sparse matrix rather than a sliding window as in those methods. Our modification to relative position encoding also makes our model invariant to the ordering of references, helping it generalize to different editing orders at inference time.

## A.3  CODE COMPLETION EXAMPLES

To help the reader see why including contextual changes can be beneficial for (editing-related) code completion problems, we compare Coeditor and InCoder6B's outputs on 3 example problems from our test set in the next few pages (Figure 5–Figure 10). These examples are sampled from a subset that are small enough to be presentable within one or two pages and in which Coeditor outperforms InCoder6B.

## A.4 MULTI-ROUND EDITING EXAMPLES

We show 3 multi-round editing examples from our test set in Figure 11–Figure 15. These examples are sampled from a subset that are small enough to be presentable within two pages and in which Coeditor achieved 50–100 total keystrokes edit gain.

```
--------------------------------------------------------------------------------
      # module: anyio._core._testing
      def get_running_tasks() -> list[TaskInfo]:
          """

          Return a list of running tasks in the current event loop.

          :return: a list of task info objects

          """
  -        return get_asynclib().get_running_tasks()
   <infill here>

=================================
Ground Truth:
  return get_async_backend().get_running_tasks()
Coeditor Prediction:
  return get_async_backend().get_running_tasks()
Incoder Prediction:
  return get_asynclib().get_running_tasks()
```

Figure 5: Code completion example 1. Coeditor sees from the relevant contextual changes (shown in Figure 6) that some `get_asynclib()` calls should be replaced with `get_async_backend()`, so it correctly suggested the change based on the deletion before the infilling point. InCoder was not able to see the deletion and infilled the original code given only the surrounding code.

```
===========changed ref 0===========
    # module: anyio._core._testing
    def get_current_task() -> TaskInfo:
        """

        Return the current task.

        :return: a representation of the current task

        """
  +        return get_async_backend().get_current_task()
  -        return get_asynclib().get_current_task()

===========changed ref 1===========
    # module: anyio.from_thread
    class BlockingPortal:
        def __new__(cls) -> BlockingPortal:
  +            return get_async_backend().create_blocking_portal()
  -            return get_asynclib().BlockingPortal()

===========changed ref 2===========
    # module: anyio._core._eventloop
    def get_cancelled_exc_class() -> type[BaseException]:
        """Return the current async library's cancellation exception class."""
  +        return get_async_backend().cancelled_exception_class()
  -        return get_asynclib().CancelledError
```

Figure 6: Code completion example 1: relevant contexts. The changes highlighted in orange tell Coeditor that some `get_asynclib()` calls should be replaced with `get_async_backend()`.

```
    ---------------------------------------------------------------------------------
        # module: instructor.oracle_data.seqgan_instructor
        class SeqGANInstructor(BasicInstructor):
            def __init__(self, opt):
                super(SeqGANInstructor, self).__init__(opt)

                # generator, discriminator
                self.gen = SeqGAN_G(cfg.gen_embed_dim, cfg.gen_hidden_dim,
                                    cfg.vocab_size, cfg.max_seq_len,
                                    cfg.padding_idx, cfg.temperature, gpu=cfg.CUDA)
                self.dis = SeqGAN_D(cfg.dis_embed_dim, cfg.vocab_size,
                                    cfg.padding_idx, gpu=cfg.CUDA)
                self.init_model()

                # Optimizer
                self.gen_opt = optim.Adam(self.gen.parameters(), lr=cfg.gen_lr)
                self.dis_opt = optim.Adam(self.dis.parameters(), lr=cfg.dis_lr)

                # Criterion
                self.mle_criterion = nn.NLLLoss()
                self.dis_criterion = nn.CrossEntropyLoss()

                # DataLoader
    +           self.oracle_samples = torch.load(cfg.oracle_samples_path)
        <infill here>
                self.gen_data = GenDataIter(self.gen.sample(cfg.batch_size, cfg.batch_size))

    =================================
Ground Truth:
    self.oracle_data = GenDataIter(self.oracle_samples)
Coeditor Prediction:
    self.oracle_data = GenDataIter(self.oracle_samples)
Incoder Prediction:
    self.oracle = Oracle(self.oracle_samples)
```

Figure 7: Code completion example 2. Coeditor was able to suggest the correct code based on a similar change from another file (Figure 8, highlighted in orange), whereas InCoder was not able to see the change and suggested a wrong statement.

```
    ===========changed ref 19===========
        # module: instructor.oracle_data.relgan_instructor
        class RelGANInstructor(BasicInstructor):
            def __init__(self, opt):
                super(RelGANInstructor, self).__init__(opt)

                # generator, discriminator
                self.gen = RelGAN_G(cfg.mem_slots, cfg.num_heads, ↵
    cfg.head_size, cfg.gen_embed_dim, cfg.gen_hidden_dim,
                                    cfg.vocab_size, cfg.max_seq_len, ↵
    cfg.padding_idx, gpu=cfg.CUDA)
                self.dis = RelGAN_D(cfg.dis_embed_dim, cfg.max_seq_len, ↵
    cfg.num_rep, cfg.vocab_size, cfg.padding_idx,
                                    gpu=cfg.CUDA)

                self.init_model()

                # Optimizer
                self.gen_opt = optim.Adam(self.gen.parameters(), lr=cfg.gen_lr)
                self.gen_adv_opt = optim.Adam(self.gen.parameters(), lr=cfg.gen_adv_lr)
                self.dis_opt = optim.Adam(self.dis.parameters(), lr=cfg.dis_lr)

                # Criterion
                self.mle_criterion = nn.NLLLoss()

                # DataLoader
    +           self.oracle_samples = torch.load(cfg.oracle_samples_path)
    +           self.oracle_data = GenDataIter(self.oracle_samples)
                self.gen_data = GenDataIter(self.gen.sample(cfg.batch_size, cfg.batch_size))
```

Figure 8: Code completion example 2: relevant contexts.

```
--------------------------------------------------------------------------------
def get_config(dataset: str,
               num_states: Optional[int] = None,
               shots: Optional[int] = 0,
               with_bow: Optional[bool] = True,
               encoder_embedding_type: Optional[str] = model_config.GLOVE_EMBED,
               decoder_embedding_type: Optional[str] = model_config.GLOVE_EMBED,
               shared_embedding: Optional[bool] = False,
               bert_embedding_type: Optional[str] = 'base',
               bert_dir: Optional[str] = '') -> config_dict.ConfigDict:
  """Returns the configuration for this experiment.
  […]
  """
  config = config_dict.ConfigDict()
  config_dir = get_config_dir(dataset)

  # config.max_per_task_failures = -1
  # config.max_task_failures = 10

  config.platform = 'jf'
  config.tpu_topology = '2x2'

  config.seed = 8

  config.dataset = dataset
  config.dataset_dir = data_utils.get_dataset_dir(dataset)
<infill here>
  config.train_epochs = 10
  config.train_batch_size = 16
  config.eval_batch_size = 16
  # Batch size for inference. Predicting in batches in case of OOM.
  config.inference_batch_size = 300
  # Seed used to generate datasets for inference.
  config.inference_seed = 9527
  # Directory storing the saved model and model prediction outputs.
  config.model_base_dir = None
  # Directory or checkpoint to initalize the model from. The initialize priority
  # will be:
  # -init_checkpoint
  # -latest checkpoint in init_dir
  # -latest checkpoint in output_dir
  config.init_checkpoint = None
  config.init_dir = None
  […]

===================================
Ground Truth:
  config.domain_adaptation = False
Coeditor Prediction:
  config.domain_adaptation = False
Incoder Prediction:
  config.train_split = 'train'
```

Figure 9: Code completion example 3. Coeditor was able to suggest adding the correct attribute initialization based on the new usage highlighted in Figure 10, whereas InCoder was not able to see the new usages and hallucinated a new attribute.

```
===========changed ref 15===========
    # module: experimental.language_structure.vrnn.train
    def run_experiment(config: config_dict.ConfigDict, output_dir: str):
    # offset: 1
    model_dir, 'labeled_dialog_turn_ids.txt'), 'w') as f:
            f.write('\n'.join(
                str(id) for id in labeled_dialog_turn_ids.numpy().tolist()))
      else:
        labeled_dialog_turn_ids = None

  +    if config.domain_adaptation:
  +      inputs = preprocessor.get_full_dataset_outputs(train_dataset_builder)
  +      # Notice domain label id 0 is also treated as in-domain: ood should have
  +      # a different id from it.
  +      in_domains, _ = tf.unique(tf.reshape(inputs[_DOMAIN_LABEL_NAME], [-1]))
  +      metric_namespaces = [
  +          _metric_namespace(_TRAIN),
  +          _metric_namespace(_TEST, True),
  +          _metric_namespace(_TEST, False)
  +      ]
  +      fewshot_metric_namespaces = [
  +          _metric_namespace(_FEWSHOT_NAMESPACE, True),
  +          _metric_namespace(_FEWSHOT_NAMESPACE, False)
  +      ]
  +    else:
  +      in_domains = None
  +      metric_namespaces = [_metric_namespace(split) for split in _SPLITS]
  +      fewshot_metric_namespaces = [_metric_namespace(_FEWSHOT_NAMESPACE)]
  +
  +    data_preprocessor = _build_data_processor(config, labeled_dialog_turn_ids,
  +                                              in_domains)
  -    data_preprocessor = _build_data_processor(config, labeled_dialog_turn_ids)
      preprocess_fn = data_preprocessor.create_feature_and_label

      # Load PSL configs
      psl_learning = config.psl_constraint_learning_weight > 0
      psl_inference = config.psl_constraint_inference_weight > 0
      if psl_learning or psl_inference:
        with tf.io.gfile.GFile(
            config.model.vae_cell.decoder_embedding.vocab_file_path, 'r') as f:
          vocab = f.read()[:-1].split('\n')
        preprocess_fn = psl_utils.psl_feature_mixin(preprocess_fn, config.dataset,
                                                    config.psl, vocab)

      # Load datasets
      # TODO(yquan): invesigate why distributed training fails in *fish TPU
      # Failure example: https://xm2a.corp.google.com/experiments/33275459
      distributed_training = False
      train_dataset = preprocessor.create_dataset(train_dataset_builder,
```

Figure 10: Code completion example 3: relevant contexts.

```
project: google/uncertainty-baselines
commit: '3e74384539 Adding support for tb.dev h…'
path: uncertainty_baselines.experiments.deterministic.eval/run_eval_epoch
{'n_references': 6, 'changed_reference_tks': 3247, 'unchanged_reference_tks': 37}
-------------------------------------------------------------------------------
editing round: 1
========Ground Truth========
 <1>:    +    val_outputs_np = None
 <5>:    +        if hparams:
         +            hp.hparams(hparams)
<13>:    +      if hparams:
         +          hp.hparams(hparams)
<15>:    +    return val_outputs_np, {k: v.numpy() for k, v in test_outputs.items()}

========Main Code========
       baselines.experiments.deterministic.eval
       def run_eval_epoch(
           val_fn: EvalStepFn,
           val_dataset: tf.data.Dataset,
           val_summary_writer: tf.summary.SummaryWriter,
           test_fn: EvalStepFn,
           test_dataset: tf.data.Dataset,
           test_summary_writer: tf.summary.SummaryWriter,
    +      current_step: int,
    -      current_step: int):
    +      hparams: Optional[Dict[str, Any]]):
 <0>     """Run one evaluation epoch on the test and optionally validation splits."""
 <1>     if val_dataset:
 <2>       val_iterator = iter(val_dataset)
 <3>       val_outputs = val_fn(val_iterator)
 <4>       with val_summary_writer.as_default():
 <5>         for name, metric in val_outputs.items():
 <6>           tf.summary.scalar(name, metric, step=current_step)
 <7>       val_outputs_np = {k: v.numpy() for k, v in val_outputs.items()}
 <8>       logging.info(
 <9>           'Validation metrics for step %d: %s', current_step, val_outputs_np)
<10>     test_iterator = iter(test_dataset)
<11>     test_outputs = test_fn(test_iterator)
<12>     with test_summary_writer.as_default():
<13>       for name, metric in test_outputs.items():
<14>         tf.summary.scalar(name, metric, step=current_step)
<15>

========Predicted Changes========
 <5>:    +        if hparams:
         +            hp.hparams(hparams)
<13>:    +      if hparams:
         +          hp.hparams(hparams)

========Accepted gains========
keystrokes gain: 94
diff-lines gain: 4
levenshtein gain: 88
```

Figure 11: Multi-round editing example 1. Coeditor correctly suggested a subset of the ground-truth changes. Contextual changes omitted for this example.

```
project: facebookresearch~pytorchvideo
commit: 'b0ae784244 Get audio samples only from…'
path: pytorchvideo.data.video/VideoPathHandler.video_from_path
{'n_references': 1, 'changed_reference_tks': 0, 'unchanged_reference_tks': 167}
--------------------------------------------------------------------------------
editing round: 3
=======Ground Truth=======
 <0>:    +                    decode_video=decode_video,
         +                    decode_audio=decode_audio,
         +                    decoder=decoder,
         +                )
         -                return EncodedVideo.from_path(filepath, decode_audio, decoder)

=======Main Code=======
      # module: pytorchvideo.data.video
      class VideoPathHandler(object):
     +     def video_from_path(
     +         self, filepath, decode_video=True, decode_audio=False, decoder="pyav", fps=30
     +     ):
     -     def video_from_path(self, filepath, decode_audio=False, decoder="pyav", fps=30):
             try:
                 is_file = g_pathmgr.isfile(filepath)
                 is_dir = g_pathmgr.isdir(filepath)
             except NotImplementedError:

                 # Not all PathManager handlers support is{file,dir} functions,
                 # when this is the case, we default to assuming the path is a file.
                 is_file = True
                 is_dir = False

             if is_file:
                 from pytorchvideo.data.encoded_video import EncodedVideo

     +           return EncodedVideo.from_path(
     +               filepath,
<0>              return EncodedVideo.from_path(filepath, decode_audio, decoder)
<1>          elif is_dir:
<2>              from pytorchvideo.data.frame_video import FrameVideo
<3>
<4>              assert not decode_audio, "decode_audio must be False when using FrameVideo"
<5>              return FrameVideo.from_directory(
<6>                  filepath, fps, path_order_cache=self.path_order_cache
<7>              )
<8>          else:
<9>              raise FileNotFoundError(f"{filepath} not found.")
<10>

=======Predicted Changes=======
 <0>:    +                    decode_audio,
         +                    decoder,
         +                    fps,
         +                    video_frame_to_path_fn=self.video_frame_to_path_fn,
         +                )
         -                return EncodedVideo.from_path(filepath, decode_audio, decoder)
```

Figure 12: Multi-round editing example 2 (round 3). Coeditor misunderstood the user's intention and suggested adding two more arguments to the `EncodedVideo.from_path` function call. Under our multi-round evaluation strategy, we assume the user would then manually add the next line from the ground truth changes (see the next figure).

```
editing round:4
========Ground Truth========
 <0>:    +                    decode_audio=decode_audio,
         +                    decoder=decoder,
         +                )
         -                return EncodedVideo.from_path(filepath, decode_audio, decoder)

========Main Code========
        # module: pytorchvideo.data.video
        class VideoPathHandler(object):
    +       def video_from_path(
    +           self, filepath, decode_video=True, decode_audio=False, decoder="pyav", fps=30
    +       ):
    -       def video_from_path(self, filepath, decode_audio=False, decoder="pyav", fps=30):
            try:
                is_file = g_pathmgr.isfile(filepath)
                is_dir = g_pathmgr.isdir(filepath)
            except NotImplementedError:

                # Not all PathManager handlers support is{file,dir} functions, when this is the
                # case, we default to assuming the path is a file.
                is_file = True
                is_dir = False

            if is_file:
                from pytorchvideo.data.encoded_video import EncodedVideo

    +           return EncodedVideo.from_path(
    +               filepath,
    +               decode_video=decode_video,
 <0>            return EncodedVideo.from_path(filepath, decode_audio, decoder)
 <1>        elif is_dir:
 <2>            from pytorchvideo.data.frame_video import FrameVideo
 <3>
 <4>            assert not decode_audio, "decode_audio must be False when using FrameVideo"
 <5>            return FrameVideo.from_directory(
 <6>                filepath, fps, path_order_cache=self.path_order_cache
 <7>            )
 <8>        else:
 <9>            raise FileNotFoundError(f"{filepath} not found.")
<10>

========Predicted Changes========
 <0>:    +                    decode_audio=decode_audio,
         +                    decoder=decoder,
         +                )
         -                return EncodedVideo.from_path(filepath, decode_audio, decoder)
========Accepted gains========
keystrokes gain: 50
diff-lines gain: 4
levenshtein gain: 47
```

Figure 13: Multi-round editing example 2 (round 4). With the next line change from the ground truth added, Coeditor understood that the user intended to only change the calling style and was thus able to suggest the correct change.

```
project: qiandao-today~qiandao
commit: [8d0f0e04df Cursor操作结束后自动关闭；统一DB连接操作]
path: db.basedb/BaseDB._insert
{'n_references': 5, 'changed_reference_tks': 957, 'unchanged_reference_tks': 168}
-----------------------------------------------------------------------------
editing round: 1
========Ground Truth========
<13>:    +           lastrowid = dbcur.lastrowid
         +           dbcur.close()
         +           return lastrowid
         -           return dbcur.lastrowid

========Main Code========
      # module: db.basedb

      class BaseDB(object):

          def _insert(self, tablename=None, **values):
 <0>          tablename = self.escape(tablename or self.__tablename__)
 <1>          if values:
 <2>              _keys = ", ".join((self.escape(k) for k in values.keys()))
 <3>              _values = ", ".join([self.placeholder, ] * len(values))
 <4>              sql_query = "INSERT INTO %s (%s) VALUES (%s)" % (tablename, _keys, _values)
 <5>          else:
 <6>              sql_query = "INSERT INTO %s DEFAULT VALUES" % tablename
 <7>          logger.debug("<sql: %s>", sql_query)
 <8>
 <9>          if values:
<10>              dbcur = self._execute(sql_query, list(values.values()))
<11>          else:
<12>              dbcur = self._execute(sql_query)
<13>          return dbcur.lastrowid
<14>

========Predicted Changes========
<13>:    +           lastrowid = dbcur.lastrowid
         +           dbcur.close()
         +           return lastrowid
         -           return dbcur.lastrowid

========Accepted gains========
keystrokes: 50
diff-lines: 4
levenshtein: 47
```

Figure 14: Multi-round editing example 3. Coeditor was able to predict the correct change in the first editing round by identifying a similar change inside a different function (see Figure 15, highlighted in orange).

```
===========signatures ref 0===========
    at: db.basedb
        logger = logging.getLogger('qiandao.basedb')

    at: db.basedb.BaseDB
        placeholder = "%s" # mysql

        _execute(sql_query, values=[])
        _execute(self, sql_query, values=[])

    at: db.basedb.BaseDB._replace
        tablename = self.escape(tablename or self.__tablename__)
[More sinagures …]

===========changed ref 0===========
    # module: db.basedb

    class BaseDB(object):
  +     # placeholder = '?' # sqlite3
  +
  +     def __init__(self, host=config.mysql.host, port=config.mysql.port,
  +             database=config.mysql.database, user=config.mysql.user, passwd=↵
config.mysql.passwd, auth_plugin=config.mysql.auth_plugin):
  +         import mysql.connector
  +         self.conn = mysql.connector.connect(user=user, password=passwd, host=host, port=port,
  +                 database=database, auth_plugin=auth_plugin, autocommit=True)
  +

===========changed ref 1===========
    # module: db.basedb

    class BaseDB(object):

        def _replace(self, tablename=None, **values):
            tablename = self.escape(tablename or self.__tablename__)
            if values:
                _keys = ", ".join(self.escape(k) for k in values.keys())
                _values = ", ".join([self.placeholder, ] * len(values))
                sql_query = "REPLACE INTO %s (%s) VALUES (%s)" % (tablename, _keys, _values)
            else:
                sql_query = "REPLACE INTO %s DEFAULT VALUES" % tablename
            logger.debug("<sql: %s>", sql_query)

            if values:
                dbcur = self._execute(sql_query, list(values.values()))
            else:
                dbcur = self._execute(sql_query)
  +         lastrowid = dbcur.lastrowid
  +         dbcur.close()
  +         return lastrowid
  -         return dbcur.lastrowid
[More changed references…]
```

Figure 15: Multi-round editing example 3 (reference blocks). The bottom changes highlighted in orange are similar to the changes needed in Figure 14.

