# OpenReview forum: "Coeditor: Leveraging Repo-level Diffs for Code Auto-editing"
_ICLR.cc/2024/Conference — ICLR 2024 spotlight_

### Official Review · Reviewer_vqdF · 2023-10-31

**Soundness:** 2 fair
**Presentation:** 4 excellent
**Contribution:** 2 fair
**Rating:** 5
**Confidence:** 5

**Summary:**

Language models effectively model text generation in a single context, but programs are often written by altering code
across many files and locations. This work proposes to fine-tune a language model with a dataset of code "diffs", consisting of a series of edits that together comprise a meaningful change. Specifically, it introduces a block-sparse attention mechanism that allows the model to attend to many related changes without at a fairly modest cost and introduces "signatures" of used variables and functions. The resulting model offers strong results at single-line and multi-line prediction given previous edits compared to language models that are unable to leverage such information.

**Strengths:**

Utilizing recent edits in a language model trained on code is a natural choice and this paper offers a reasonable implementation of this idea. Compared to other recent work with the same general goal, it is particularly strong in its definition of reference block and the corresponding sparse attention pattern. This allows it to scale to fairly long contexts, which yields a modest boost in performance (Tab. 5).

While the results come with some concerns (see below), the "replace" performance in particular is quite promising. The work is also quite well written. Overall, this paper presents a promising exploration of its domain.

**Weaknesses:**

The evaluation raises some concerns, in particular the comparison with the baseline models. This work introduces a few components; the key one is the notion of reference blocks that CodeT5 can attend to when responding to a query. The baseline models do not have access to these reference blocks since they are trained on regular token streams. This creates a few issues, some of which can be resolved with clarifications in the writing and others that might require further experiments. In no particular order:

- There can be many reference blocks, spanning up to 16.4K tokens, so the work uses a relatively short "Query" of up to 1,024 tokens. Other LMs, such as InCoder and StarCoder, all have much larger context sizes. It is not clear from the writing whether the other models were "allowed" to use their full context window, or were provided with the same limited section of context. If a completion took place halfway through an 8K token file, one would expect StarCoder, for instance, to do substantially better with the full file context than with a 1K token window.
- A seemingly unrelated form of "reference" comes from the "signatures" (section 3.2). This idea does not have anything to do with code editing itself, but rather with providing an LM access to the project-level context. This seems to be very impactful (Tab. 5). It looks like the baseline LMs were not provided with this information, which seems like an oversight. For one, there are reasonable ways to prompt pretrained language models with such information even if that wasn't present at pretraining time. For another, the performance of this component has little, if anything, to do with the stated goal of the work (leveraging diffs for edit prediction). The fact that it boosts performance so strongly makes the results much less interesting; it suggests that a large component of the performance does not come from the presence of diffs, but from a more banal form of information that related work has already explored. The paper should either include carefully calibrated experiments with baselines using the same information or exclude this from its main contributions in results such as Tab. 3.
- Relatedly, it is not clear why there is such a large performance gap between Tab. 5 and Tab 3. The former uses the validation set, so one might expect that the latter is based on test data. The text in 4.3 does mention that the model was only halfway trained here. But the gap between 42.1% and 60.4% is really very large. Is this not the same type of task as in Tab. 3? And how should we expect the gaps between ablations to scale to the numbers of Tab. 3? In particular, would the impact of the retrieval blocks (criticized above) be proportional or even disproportionally larger?
- The lack of a comparison between models in Sec. 4.2/Tab. 4 is also somewhat surprising and concerning. Please endeavour to replicate these numbers with at least one strong baseline, e.g. StarCoder.

On a minor note, it was quite surprising to see it mentioned that the model was trained with a batch size of 1 (P7). Was gradient accumulation considered for increasing the batch size? That might improve performance.

Minor issues:
- P5: "making the model is" -> remove "is"
- P6: "statictics" -> "statistics"

**Questions:**

Based on the issues noted above:
- Were the baseline LMs evaluated with a 1K context, same as Coeditor, or with their maximum number of tokens (when possible)?
- Was signature information only used in Coeditor? If so, is this intended to be a core part of the contribution? Please provide ablations with other models using this information, and with Coeditor fully trained without this information in any case.
- How do the baseline models perform in the multi-round setting?

---

> ### Author Response · Authors · 2023-11-15
> **Response to Reviewer vqdF**
>
> Thank you for the review and questions. Please see our response below:
>
> __Q1: Were the baseline LMs evaluated with a 1K context, same as Coeditor, or with their maximum number of tokens (when possible)?__
>
> We let all baseline LMs utilize their full context by including as much surrounding code as possible. The context size of each model is detailed in Table 3.
>
> ---
>
> __Q2: Was signature information only used in Coeditor? If so, is this intended to be a core part of the contribution? Please provide ablations with other models using this information, and with Coeditor fully trained without this information in any case.__
>
> We would first like to clarify that all our ablation results were obtained by retraining Coeditor with the corresponding information removed at training time (as mentioned at the beginning of Section 4.3.)
>
> The signature information (based on usage-analysis) is only used by Coeditor in our evaluation and is part of the contribution of this work. While prior work has explored the use of signatures in tasks like code completion [1] and type inference [2], their value in code editing has not been studied before. In fact, our initial hypothesis was that signatures might be less critical in editing tasks, given that the model has access to the old version of the code, which can aid in inferring signatures from existing usages.
>
> However, our ablation studies (Table 5) uncovered a significant finding: both recent editing history and signature information are crucial for editing performance, and neither alone is sufficient. In particular, the first row of Table 5 shows that not having editing history (while still keeping the signatures and a 16K context size) leads to the largest drop in performance, reducing overall EM from 42.1% to 26.1%. This result counters the statement that “a large component of the performance does not come from the presence of diffs, but from a more banal form of information that related work has already explored”.
>
> To understand why the combination of signatures and recent editing history is so useful for this task, we manually inspected examples where Coeditor was successful and found that the editing history often clarifies the desired types of changes, while the signature information aids in identifying where similar changes should be applied. For example, if the model sees that the user has added a statement `obj.some_attribute *= 2`  and needs to predict a similar change `obj.another_attribute *= 2` ,  it often requires seeing what other attributes are defined in `obj`’s class signature in order for the model to predict which other attributes need to be similarly mutated.
>
> [1] Toufique Ahmed et al. "Improving Few-Shot Prompts with Relevant Static Analysis Products." arXiv preprint arXiv:2304.06815 (2023).
>
> [2] Jiayi Wei et al. "TypeT5: Seq2seq Type Inference using Static Analysis." The Eleventh International Conference on Learning Representations. 2023.
>
> ---
>
> __Q3: How do the baseline models perform in the multi-round setting?__
>
> The intent of the multi-round setting is to evaluate how the model performs when iteratively prompted with additional user changes (if the model didn’t predict all the needed changes in the first round). Since the code infilling baselines cannot predict where and how to make edits, we had to evaluate them on a simplified code completion task. This adaptation means that we cannot meaningfully apply the multi-round editing framework to these models.
>
> For example, if the ground truth is
>
> ```
> -def sort_by_name(persons):
> +def sort_by_name(persons, reversed=False):
>      persons.sort(key=lambda p: p.name)
> +    if reversed:
> +        persons.reverse()
>      return persons
> ```
>
> We create the corresponding code completion problem below, where we only select the last changed line as the code completion target and apply the remaining code changes to form the new context. This creates a normal completion problem such that the resulting code (after adding the missing line) is the after-commit version of the code above.
>
> ```
> def sort_by_name(persons, reversed=False):
>     persons.sort(key=lambda p: p.name)
>     if reversed:
>         <target> persons.reverse() </target>
>     return persons
> ```
>
> This is similar to the last round in a multi-round editing setting, arguably the *easiest part* of a multi-round editing loop for the baseline models, which were not pre-trained to handle a partially edited context.

---

> > ### Comment · Reviewer_vqdF · 2023-11-22
> >
> > Thanks for your detailed response. Good to hear that context length was not a confound. Regarding the multi-round editing task: I understand that the baseline models are not particularly compatible to this task, but I mainly brought this up because that section introduces a new, and interesting, metric. The significance of this metric and of Coeditor's potential productivity gains quantified by it, are hard to interpret from results involving just the proposed model. It helps to provide baselines, even if poorly suitable to the task, to help ground whether a measurement like 28% keystrokes saved should be interpreted as moderately useful, amazing, or something else. I think it's not a stretch to come up with an evaluation that uses traditional models in a similar way. In any case, I'll leave this up to you.
> >
> > Regarding the signatures, I think the response misses the key point I made in my review. The problem isn't that using signatures is wrong -- as you also mention, they are quite commonly used -- or that I think they matter more than edit history -- Table 5 clearly contradicts this. It's that signatures matter quite a lot -- performance drops by some 20% when they are removed, more than half the drop recorded for "no diffs" -- and that this information is not made accessible to the other models evaluated in this work. This somewhat obfuscates the finding that diffs alone provide somewhat limited performance gains and do not explain the full performance gap reported in Tab. 3. The title/abstract/intro/conclusion also make little mention of signatures being a "crucial" component of Coeditor's performance. It is important to acknowledge this in writing, both noting that diff information alone is often insufficient in the aforementioned sections, and either adding more equal comparisons with baselines or noting the absence of signature information in baseline models as a limitation of the evaluation.
> >
> > On a minor note: I would still appreciate clarification on why Tab. 5 shows a best-case EM rate of 42.1% while Tab. 3 shows a much higher value of 60.4%. Could you comment on this? It seems like a very large gap even if the two are on different datasets (the dev and test set respectively); the two would have to be distributed quite differently.

---

> ### Author Response · Authors · 2023-11-22
>
> __Q: I would still appreciate clarification on why Tab. 5 shows a best-case EM rate of 42.1% while Tab. 3 shows a much higher value of 60.4%.__
>
> This is because these two tables show results on different tasks. Tab. 3 shows the EM rate on the PYCOMMITS-ONELINE dataset, which is a code completion task involving only one missing line. Tab. 5 shows the single-round EM rate on PYCOMMITS, which is an editing task requiring identifying and performing multiple missing changes. Because the latter is a fundamentally harder problem, the EM rate is lower.
>
>
>
> __Comment 1: Regarding the multi-round editing task: I understand that the baseline models are not particularly compatible to this task [...] I think it's not a stretch to come up with an evaluation that uses traditional models in a similar way. In any case, I'll leave this up to you.__
>
> The fundamental difficulty of the multi-round editing task is that it requires predicting more than one change, hence if a model misses some changes in the first round, it can potentially benefit from additional changes made by the user. Code completion models like StarCoder are only trained to predict a single missing expression (via the Fill-in-the-Middle objective), so we think there is no meaningful way to evaluate such models on the PYCOMMITS dataset in a way that’s comparable to our model. In any case, we’ll consider this more deeply for future work; thanks for the suggestion!
>
> __Comment 2: It is important to acknowledge this in writing, both noting that diff information alone is often insufficient in the aforementioned sections, and either adding more equal comparisons with baselines or noting the absence of signature information in baseline models as a limitation of the evaluation.__
>
> We will ensure that both points you highlighted are addressed in our upcoming revisions.
> The decision to withhold signature information from the code completion models stemmed from their inherent limitations, as out-of-the-box, they typically struggle to utilize signatures effectively due to how they were pre-trained (context is effectively limited to a single document consisting of code rather than signatures). Consequently, it becomes a challenge to effectively prompt these models with signature information while also balancing the limited token budgets.

---

### Official Review · Reviewer_sVPE · 2023-10-31

**Soundness:** 3 good
**Presentation:** 4 excellent
**Contribution:** 4 excellent
**Rating:** 6
**Confidence:** 3

**Summary:**

The authors propose to shift the focus from learning code completion to learning edits instead and leveraging repository wide line-diff information to do so. They build on top of CodeT5 which they augment with sparse block attention to enable larger context sizes needed for the new task to produce Coeditor. Further, they create a summary "scratchpath" that summarises useful, out-of-file information, the summarisation being crutial to avoid large prompts. They derive an edit history from the git histories of Python projects and use the current edited files as context at prediction time. Further, they integrate into VSCode as an extension to demonstrate how such a tool can work in a developer workflow. Crucially, the novel setting is the human-in-the-loop, multi-round editing, where a human may provide feedback between rounds. Evaluating using Exact Match, they demonstrate better performance both in single- and multi-round editing settings, demonstrating the befit of shifting to modeling edits directly, thr ablation study further demonstrates the value of diffs.

**Strengths:**

- A perspective shift to modelling code edits.
- A smaller scale model that outperforms larger models on the code auto-completion task.
- VSCode Extension that can be used directly in a project.
- A historical edits dataset for Python

**Weaknesses:**

- The edit granularity choice and dataset can introduce bias: developers often commit many intermediate commits that get squashed into a single commit before pushing and changes can get overwritten.
- While the delta/diff encoding is justified, the space feels under-explored.

After reading the separate discussion threads, I share the concern of the reviewer vqdF regarding the use of signatures and their impact on performance. I am unconvinced that the information cannot be exposed to baselines, for example, by borrowing methodology from Ahmed et al. [^1].

[^1]: Ahmed, T., Pai, K. S., Devanbu, P., & Barr, E. T. (2023). Improving Few-Shot Prompts with Relevant Static Analysis Products. arXiv preprint arXiv:2304.06815. https://arxiv.org/pdf/2304.06815.pdf

**Questions:**

With the move towards an extension, was using more granular edit events considered, what about navigation events or other similar side-channel information? Is the main concern the context size or are there other trade-offs that moved the authors closer to git diff, line-level information?

---

> ### Author Response · Authors · 2023-11-15
> **Response to Reviewer sVPE**
>
> Thank you for the review and questions. Please see our response below:
>
> __Comment: The edit granularity choice and dataset can introduce bias: developers often commit many intermediate commits that get squashed into a single commit before pushing and changes can get overwritten.__
>
> This is a good point: in practice, commits in a pull request are often squashed before merging, resulting in training commits that are larger than what users typically encounter at editing time. There are several potential remedies to this issue. For example, at inference time, we could extend our approach to include editing history beyond the last commit, in order to better match the content of a squashed commit. Future work can also explore using LLMs to identify squashed commits and decompose them into smaller, semantically independent units. Notably, however, many squashed commits collected from open source repos can still be relatively small and contain closely related changes. Therefore, training our model on a substantial volume of such commits may already provide a robust training signal that is sufficient for teaching the model to utilize editing histories.
>
>
> __Question: With the move towards an extension, was using more granular edit events considered, what about navigation events or other similar side-channel information? Is the main concern the context size or are there other trade-offs that moved the authors closer to git diff, line-level information?__
>
> Our choice of using commit history is mainly motivated by data availability. Acquiring detailed user edit events entails installing specialized tools for data collection, posing scalability and privacy challenges that are beyond the scope of this work. A promising future direction, however, could involve a hybrid approach that combines large-scale pre-training (using the commit-based data proposed by this work) followed by fine-tuning with a smaller dataset of detailed edit events.

---

> > ### Comment · Reviewer_sVPE · 2023-11-17
> > **Follow up on Q1**
> >
> > Thank you for the answers, and I tend to mostly agree, in particular, I agree that this falls just beyond the scope of this paper and perhaps ICRL as this would make the paper more SE-focused.
> >
> > The concern I had with the bias, and the reason Q2 focused more on IDE edit events is that the type of edits that will not be visible in the squashed commit are those where overwriting happens. Perhaps ironically, these shadowed diffs are those I would speculate are good candidates to learn to predict since they are exactly those edits that developers had to do to make progress on the task. Sadly, I will concede the data collection issue, especially in a privacy-preserving way.

---

### Official Review · Reviewer_Kf1h · 2023-10-31

**Soundness:** 3 good
**Presentation:** 3 good
**Contribution:** 3 good
**Rating:** 8
**Confidence:** 4

**Summary:**

This paper introduces Coeditor, a novel model for auto-editing code based on the CodeT5 transformer model architecture. It is designed to predict edits to existing code, based on the related recent changes in the same codebase. The tool requires users to manually select the code region for editing. The paper introduces a new PYCOMMITS dataset, sourced from 1650 open-source Python projects, which is used for evaluation. The model outperforms existing tools in both single-edit and multi-round editing scenarios and includes a VSCode extension for practical use. The paper highlights the importance of integrating editing history in code completion and provides resources for future research in this domain.

**Strengths:**

* Introduce and open source the new research PyCommits dataset, source code, and VSCode extension. Which is an important contribution to future research in this domain.
* A step towards solving an important practical problem, and a potential to integrate in real-world developer tools

**Weaknesses:**

* One notable weakness of the Coeditor model is its reliance on users to manually pinpoint the regions in the code that need editing (as pointed by the authors, too). This approach limits the model's potential for broader applications, such as automated refactoring, and places additional steps in the workflow that could be automated for enhanced efficiency and user experience.

* The evaluation section could be improved by incorporating a more diverse range of baselines. Currently, it predominantly features large language models (LLMs) tailored for coding tasks.

**Questions:**

It would strengthen the paper if you could include any specialized code editing models or tools to the evaluation, which could provide a more comprehensive comparative analysis.

I would be curious to see how CodePlan + Coeditor perform on the PyCommits dataset (especially since the CodePlan paper partially features the results using the Coeditor model), although I realized that CodePlan was published after the ICLR submission deadline. Is it possible to add?

Another interesting baseline could be BluePencil model focusing on repetitive edits (see "On the fly synthesis of edit suggestions" by Miltner et. al) this is also available through IntelliCode Suggestions VS extension.

---

> ### Author Response · Authors · 2023-11-15
> **Response to Reviewer Kf1h**
>
> Thank you for the review and questions. Please see our response below:
>
> __Comment: I would be curious to see how CodePlan + Coeditor perform on the PyCommits dataset (especially since the CodePlan paper partially features the results using the Coeditor model), although I realized that CodePlan was published after the ICLR submission deadline. Is it possible to add?__
>
> Regarding combining CodePlan with Coeditor for the PyCommits dataset, it's important to note CodePlan and Coditor’s different focuses. Coeditor, along with the PyCommits dataset, focuses on making precise changes within specific code regions. CodePlan focuses on orchestrating large-scope changes across multiple files, based on some clear specification and utilizing a local editing model similar to Coeditor. This difference in their operational scopes means PyCommits would not be an ideal evaluation for CodePlan + Coeditor, as CodePlan’s planning strength is not needed for PyCommits' localized edits.
>
> ---
>
> __Comment: Another interesting baseline could be BluePencil model focusing on repetitive edits (see "On the fly synthesis of edit suggestions" by Miltner et. al) this is also available through IntelliCode Suggestions VS extension.__
>
> BluePencil and Coeditor share a common goal of predicting user edits based on recent user changes. However, BluePencil uses rule-based program synthesis techniques and is limited to suggesting repetitive changes. In contrast, Coeditor has the capability to produce novel changes beyond BluePencil's scope. Furthermore, BluePencil's implementation is specific to C# programs, while Coeditor has been trained exclusively for Python. This difference in target programming language precludes a direct comparison of the two tools.

---

### Official Review · Reviewer_ueao · 2023-10-31

**Soundness:** 3 good
**Presentation:** 3 good
**Contribution:** 3 good
**Rating:** 6
**Confidence:** 4

**Summary:**

The paper presents Coeditor, a fine-tuned language model that is designed for code editing tasks. The backbone of the Coeditor is the CodeT5 model, which the authors have finetuned on long context (2k->4k->8k) using block-sparse attention and capitalizing the relative positional encoding scheme of CodeT5 model. Coeditor is built based on two key ideas: (1) encode prior code edits using a line-based diff scheme and decode the edits using the masked span infilling objective; and (2) using lightweight static analysis to pull in relevant parts of the codebase (e.g., function signature). The paper proposed a dataset called PyCommits which is collected from 1650 open-source Python projects on Github. The paper compared Coeditor with code infilling models - InCoder, Starcoder, text-davinci-003 and showed that Coeditor outperforms them by a large margin. Codeditor is released with code, dataset, model checkpoint, and a VSCode extension.

**Strengths:**

- A good dataset (PyCommits) that will foster future research.
- The proposed method to train sequence-to-sequence language models for code editing is sound. Overall, the writing is good (though there are minor grammar issues and repeated words) and the paper is easy to read and follow.
- An IDE extension that researchers will be able to use and understand the effectiveness of the approach.

**Weaknesses:**

The primary issue of the paper is over claim when it compares with SOTA code infilling models. Statements such as "our method achieves 60.4% exact match accuracy using a 220M parameter model, whereas text-davinci-003—the best performing code infilling model we have compared, which has 175 billion parameters—achieves only 30.5%." is extremely misleading. Coeditor is a finetuned model that is made by specializing on the task at hand. On the other hand, the infill models are general purpose models. The paper didn't explain clearly how the generic infill models are prompted to solve the code editing tasks. Therefore, there is no way we can fairly compare the proposed approach with the baseline models compared in this work.

**Questions:**

- Why Coeditor is trained with a fixed batch size of 1? The Nvidia GPU used to train the model has 48GB memory which should be good to accommodate batch size > 1 with a 220M param model, right?
- Does the PyCommits dataset composed of Github projects that have permissive licenses?
- How the generic infill models are prompted to solve the code editing tasks? Is few-shot prompting used for the models? For example, [1] used demonstration augmented prompting for Codex model (when finetuning is not possible).
- Can we use instruction finetuned version of the code generation models for code editing task?
- Instead of a seq2seq model, can we use a decoder-only LM for the editing task?
- Is it possible to evaluate Coeditor on the PIE benchmark [1]?

[1] Learning Performance Improving Code Edits (https://pie4perf.com)

---

> ### Author Response · Authors · 2023-11-15
> **Response to Reviewer ueao**
>
> Thank you for the review and questions. Please see our response below:
>
> __Comment: The primary issue of the paper is over claim when it compares with SOTA code infilling models […] The paper didn't explain clearly how the generic infill models are prompted to solve the code editing tasks. Therefore, there is no way we can fairly compare the proposed approach with the baseline models compared in this work.__
>
> Our comparison with infill models is detailed in Section 4. We extracted single-line code completion problems from real-world edits and fed the prefix and suffix of each problem to the infill model, assessing whether it can accurately output the correct middle span. We discuss other prompting strategies below under Q3.
>
> Note that the single-round editing evaluation has a structure that is designed to be fair to infilling models as follows: if the ground truth changes in a code unit $u$ contain multiple changed lines, we only select the last changed line as the code completion target and remove it from $u$, and we apply the remaining code changes to $u$. This creates a normal completion problem such that the resulting code (after adding the missing line) is the after-commit version of $u$.
> For example, if the ground truth is
> ```
> -def sort_by_name(persons):
> +def sort_by_name(persons, reversed=False):
>      persons.sort(key=lambda p: p.name)
> +    if reversed:
> +        persons.reverse()
>      return persons
> ```
> We create the corresponding code completion problem as
> ```
> def sort_by_name(persons, reversed=False):
>     persons.sort(key=lambda p: p.name)
>     if reversed:
>         <target> persons.reverse() </target>
>     return persons
>
> ```
> This is similar to the last round in a multi-round editing setting, arguably the *easiest part* of a multi-round editing loop for the baseline models, which were not pre-trained to handle a partially edited context.
>
> ---
> __Q1: Why Coeditor is trained with a fixed batch size of 1? The Nvidia GPU used to train the model has 48GB memory which should be good to accommodate batch size > 1 with a 220M param model, right?__
>
> In our training setup, we utilize about 20% of the available GPU memory, achieving roughly 70% average GPU utilization. While a larger batch size is feasible, the block-sparse attention pattern (which varies for each training example) coupled with the diverse context sizes of different editing problems (ranging from no reference blocks to 15K reference block tokens), limited the benefits of larger batch sizes for our task.
>
> ---
> __Q2: Does the PyCommits dataset composed of Github projects that have permissive licenses?__
>
> Yes, as stated in Section 3.4's last paragraph, we constructed the PyCommits dataset from the commit history of 1,650 Python projects with permissive licenses (MIT, Apache, BSD) sourced from GitHub.
>
> ---
> __Q3: How the generic infill models are prompted to solve the code editing tasks? Is few-shot prompting used for the models? For example, [1] used demonstration augmented prompting for Codex model (when finetuning is not possible).__
>
> Given that the single-line code completion task is closely related to the original infilling task, we did not employ few-shot prompting for the text-davinci-003 model. Instead, we utilized the maximum available context size (4K tokens) to store as much surrounding code as feasible.  The reported performance of text-davinci-003 was obtained by directly calling OpenAI’s infilling API, which expects a prefix and a suffix.
>
> We did experiment with different prompt formats on GPT-3.5-turbo, as described below (in the next comment).

---

> ### Author Response · Authors · 2023-11-15
> **Response to Reviewer ueao, continued**
>
> __GPT-3.5-turbo experiments__
>
> Since GPT3.5 is not a code infilling model, we tried different ways to prompt it for this task and report two results below. The first prompting method, called “GPT3.5-plain”, simply put the suffix before the prefix using the format
> ```
> {suffix code}
> —---
> {prefix code}
> ```
> and set the message role to “assistant”. This way, we encourage the model to send a reply that continues generating the remaining prefix code, and we take the first line in the reply as the model suggestion.
>
> The second prompting method we tried, “GPT3.5-instruction”, formulates a one-shot prompt. We send the following instructions to the model using the role “user”:
>
>
> ```
> You are a programming expert tasked to fill in a missing line for a given Python code
> snippet. The snippet may have been truncated from both ends, and the missing line is
> indicated by a special token `<MISSING LINE>`.
> You should output the missing line (along with any leading whitespaces) and
> nothing more. For example, if the input is
> —
> def fib(n):
>    if n < 2:
> <MISSING LINE>
>    else:
>        return fib(n-1) + fib(
> —
> Your output should be "        return 1" (without the quotes) and nothing more.
>
> Now fill in the code snippet below:
> —
> {prefix code}<MISSING LINE>{suffix code}
> —
> Your output:
> ```
>
> However, both prompting strategies performed worse than text-davinci-003, as indicated in the table below, so we did not include these results in the paper.
>
> | Model | Context Size | Add EM | Replace EM | Overall EM |
> | ---- | ---- | ---- | ---- | ---- |
> | GPT3.5-plain          | 4096 | 21.88 | 30.20 | 27.80 |
> | GPT3.5-instruction | 4096 | 6.94 | 9.55 | 8.80 |
>
> ---
>
> __Q4: Can we use instruction finetuned version of the code generation models for code editing task?__
>
> While there is recent work focusing on instruction-based editing, our approach focuses on a markedly different setting that doesn't require user instructions and is more similar to traditional code completion tools. Combining explicit developer instructions with editing history into our model could be an interesting direction for future work.
>
> ---
>
> __Q5: Instead of a seq2seq model, can we use a decoder-only LM for the editing task?__
>
> A decoder-only LM could potentially be adapted for the task presented in this work. However, some specific designs, like how T5’s mask tokens are used to model editing as masked span infilling, would likely require some adaptation for decoder-only models to work well on this task.
>
> ---
>
> __Q6: Is it possible to evaluate Coeditor on the PIE benchmark [1]?__
>
> The focus of this work is predicting probable code edits based on recent programmer changes made elsewhere, whereas the PIE benchmark focuses exclusively on performance-improving program rewrites without conditioning. Therefore, the PyCommits dataset differs fundamentally in its objectives from the PIE dataset.

---

> > ### Comment · Reviewer_ueao · 2023-11-22
> > **Thank you for answering my questions**
> >
> > Thank you for addressing my questions, I have more clarity about the work after your response. I am updating my score from 3 to 6.

---

### Meta-Review · Area_Chair_HvK7 · 2023-12-06

**Metareview:**

The paper addresses the scenario of a programmer making repeated small edits to a codebase; and using previous edits to predict the current one.  This could be useful for example in refactoring when the programmer needs to make a sequence of edits in different files which achieve a single goal.

The paper's model is evaluated on a new dataset of about 200K commits, each changing roughly 30 lines, extracted from 1650 repositories, and also provides a VSCode extension.

Strengths:
 - Reviewers engaged with the "interactive editing" paradigm, and see value in the scenario.
 - After rebuttal, reviewers are satisified with the evaluation, partly due to excellent clarifications provided by the authors in the rebuttal.

Weaknesses:
 - As agreed by the authors, there is a small mismatch between the collected data and the actual task - for example, commits may have been squashed - however reviewers agree that the dataset is useful and will spark future work.
 - Reviewers are concerned that the paper's model is given access to function and class signatures, which ablation shows to be very important to the performance of this model.  The rebuttal concedes that the use of signatures is common in the wider related work, but argues:

 >  The decision to withhold signature information from the code completion models stemmed from their inherent limitations, as out-of-the-box, they typically struggle to utilize signatures effectively due to how they were pre-trained (context is effectively limited to a single document consisting of code rather than signatures). Consequently, it becomes a challenge to effectively prompt these models with signature information while also balancing the limited token budgets.

This is not convincing - it essentially suggests that the reviewers made no attempt to supply signatures to the other models.  How are signatures not representable in code, as this implies?   Just define an empty class, or a one-line function which defers to another?  Yes, signatures will cost context space, but in the order of lines, not file-loads.  [Note this is not a new weakness introduced by the AC, I am simply reflecting the reviewers' observations.]

**Justification For Why Not Higher Score:**

The reviewers' issues with the handling of signatures, and with the potential slight mismatch of dataset and scenario mean this paper is a good spotlight, but not quite oral material.

**Justification For Why Not Lower Score:**

The paper makes a spotlight rather than a poster because the scenario is of wide interest to the ICLR audience, and the dataset construction and paper approach can appeal outside of the subfield.

---

### Decision · Program_Chairs · 2024-01-16

Accept (spotlight)